# Systematic proteomics analysis of lysine acetylation reveals critical features of renal proteins in kidney calculi formation

Shiwei Zhang[1,2]☯, Zhu Wang[2]☯, Hao Jiang[1], Jieyan Wang[2], Meiyu Jin[2], Hui Liang[2]*, Qiong Deng🆔[2]*

**1** College of Animal Sciences, Jilin University, Changchun, Jilin, China, **2** Department of Urology, The People's Hospital of Longhua, Shenzhen, Guangdong, China

☯ These authors contributed equally to this work.
* lianghui8689@smu.edu.cn (HL); dengqiong1987@smu.edu.cn (QD)

## Abstract

In this study, we systematically integrated proteome and acetyl proteome (acetylome) approaches to investigate the characteristics of renal proteins in a CaOx crystal rat model. We aimed to understand the pathogenesis of kidney calculi and delineate the landscape of acetylation within kidney calculi, potentially leading to the identification of valuable and novel biomarkers. Using liquid chromatography-tandem mass spectrometry (LC-MS/MS), we analyzed the protein expression profiles and lysine acetylation (Kac) features in kidney tissues obtained from rats with kidney calculi and those without (normal controls). Our results revealed 118 downregulated and 129 upregulated proteins. Furthermore, we identified 538 upregulated Kac sites in 258 proteins and 133 downregulated Kac sites in 118 proteins between kidney calculi and paired normal rats. Functional enrichment and protein-protein interaction network analyses revealed that the mitochondria were the most abundant acetylated protein fraction, and metabolic pathways were predominated among the GO and KEGG pathways. Furthermore, the LC-MS/MS findings were verified by immunofluorescence. The study is the first comparative study of Kac modification associated with kidney calculi. These findings offer significant insights into the molecular mechanism underlying the formation and development of renal stones.

## Introduction

Nephrolithiasis affects approximately one in 17 Chinese adults [1], with calcium oxalate (CaOx) being the predominant crystal in kidney stones [2,3]. These stone often recur at a rate exceeding 50% in five years [4], posing significant challenges for clinicians preventing their recurrence. However, kidney stone formation constitutes a complex cellular response to crystals exposure [5], with the exact mechanisms yet to be fully elucidated.

**Data availability statement:** The data generated in the present study may be found in the ProteomeXchange under accession number PXD050342 (https://www.ebi.ac.uk/pride/archive/projects/PXD050342).

**Funding:** This work was supported by the Scientific and Technological Innovation Fund of Longhua Shenzhen (grant number 11501A20240704F5EB6FF to Z.W.).

**Competing interests:** The authors have declared that no competing interests exist.

**Abbreviation:** CaOx, calcium oxalate; Kac, lysine acetylation; GO, gene ontology; SIRT3, sirtuin 3; FOXO1, forkhead box O1; VDR, vitamin D receptor; CaSR, calcium-sensing receptor; TRPV6, transient receptor potential vanilloid type 6; CYP27, cytochrome P450 24; SD, Sprague-Dawley; TCA, trichloroacetic acid; TEAB, triethyl-ammonium bicarbonate buffer; IP, immunoprecipitation; PASEF, parallel accumulation serial fragmentation; PPI, protein-protein interaction; PCA, principal component analysis; CKD, chronic kidney disease; OPN, osteopontin; MGP, matrix Gla protein; LFQ, label-free quantification; HE, hematoxylin-eosin.

Lysine acetylation (Kac) is an essential posttranslational modification first identified in histones during the 1960s [6]. A Protein-Protein Interaction (PPI) network analysis and a Gene Ontology (GO) analysis of kidney calculi formation, highlighting proteins involved in posttranslational protein modification processes [5]. Other studies similarly demonstrated that protein Kac was associated with kidney stone development. For instance, acetate, an epigenetic metabolite, has been shown to promote cancer cell survival under hypoxic stress [7] and enhance Histone H3 acetylation in renal tubular cells, consequently promoting the expression of microRNAs-130a-3p, -148b-3p and -374b-5p specifically by upregulating H3K9 and H3K27 acetylation at their promoter regions. These miRNAs can suppress the expression of Nadc1 and Cldn14, thereby enhancing urinary citrate excretion and reducing urinary calcium excretion [8]. Moreover, sirtuin 3 (SIRT3), an $NAD^+$-dependent deacetylase in the mitochondria, suppresses renal inflammation, renal apoptosis, and renal CaOx crystal formation. The inhibitory effect was mediated by the promotion of macrophage polarization toward the M2 phenotype. It also exerts anti-inflammatory and tissue-healing effects on nephropathy and ischemia/reperfusion acute kidney injury [9,10] via the deacetylation of Forkhead box O1 (FOXO1) [11]. FOXO1/ vitamin D receptor (VDR) regulator axis was reported to be involved in the calcium oxalate crystallization [12]. Similarly, VDR up or downregulates target genes, such as renal calcium-sensing receptor (CaSR), intestinal calcium transporters transient receptor potential vanilloid type 6 (TRPV6), cytochrome P450 27 (CYP27), CYP24, and VDR, associated with vitamin D metabolism and calcium homeostasis by binding to multiple promoter regions and hyperacetylating histone H3 [13]. Collectively, various studies suggest a critical role of acetylation modification in the pathogenesis of kidney stones. However, the aforementioned studies primarily focused on histone acetylation, and lacked a comprehensive understanding of the involved molecular mechanism.

This study aimed to establish a protein Kac profile using a rat model of renal calculi, since the stone formation is significant and stable in the rat model. Our research serves as a valuable resource, offering new insights into the pathogenesis of nephrolithiasis. It would be help to elucidate the mechanism involved in regulating stone formation and provide new intervention therapy targets for drug development. It is anticipated that our findings will facilitate the way for the development of new strategies and targets in the prevention and treatment of kidney stone disease, aiding in the improvement of patient care.

## Methods

### Animals

All experiments involving rats were conducted in strict compliance with the Guide for the Care and Use of Laboratory Animals, which was prepared by the Institute of Laboratory Animal Resources for the National Research Council. The study was approved by the ethics committee.

## Development of the CaOx crystal rat model

Male Sprague-Dawley (SD) rats, weighing between 225–250 g and procured from Guangdong Medical Laboratory Animal Center, were maintained for a minimum of five days under standard conditions. During this period, they were provided water and rat chow ad libitum, maintaining a 16-hour light/8-hour dark cycle. SD rats were randomly divided into a control group and kidney calculi group. We established the kidney CaOx crystal rat model following the previously reported protocol [14]. Briefly, rats (n = 3) were given free access to food and drinking water that contained 1% (v/v) ethylene glycol (EG, Sigma-Aldrich, Buchs, Switzerland) for three weeks. Additionally, each rat received 2 ml of 1% (w/v) ammonium chloride via gavage, administered on separate days, to promote the deposition of CaOx crystals. For the control group (n = 3), the rats received normal drinking water, and 2 ml of distilled water via gavage on separate days. The humane endpoints were set as following: 1) weight loss up to 20%; 2) poor appetite, with a diet 50% lower than the normal group; 3) mental depression with no response to driving, body temperature lower than 37°C. Euthanasia was performed by decapitation after $CO_2$ sedation for 1 min with a chamber flow displacement rate of $CO_2$ at around 5 liter/min (the euthanasia camber is 10 liter), reached 30–70% volume displacement rate of the $CO_2$ (NIH2020). Blood and bladder urine samples collected and promptly placed on ice. Bilateral kidneys were removed, washed in pre-cooled phosphate buffer saline (PBS) to eliminate blood contaminations, and then divided into two halves, One half was fixed in 10% neutral buffered formalin and embedded in paraffin, while the other half was rapidly frozen in liquid nitrogen for subsequent analysis.

## Label-free quantification (LFQ)-based quantitative protein acetylation analysis

**Protein extraction.** Protein extraction and quality assessment were performed following a previously established protocol [15]. The sample was grinded with liquid nitrogen into cell powder and then transferred to a 5-mL centrifuge tube. After that, four volumes of lysis buffer (8 M urea, 1% protease inhibitor cocktail, 3 μM Trichostatin A, 50 mM Nicotinamide) was added to the cell powder, followed by sonication three minutes on ice using a high intensity ultrasonic processor (Scientz). By inhibiting the activity of acyltransferases and deacylases, Trichostatin A and Nicotinamide prevent alterations in acylation modifications during the protein extraction process, thereby avoiding impacts on post-translational modifications of proteins. The remaining debris was removed by centrifugation at 12,000 g at 4°C for 10 min. Finally, the supernatant was collected and the protein concentration was determined with BCA kit according to the manufacturer's instructions.

**Trypsin digestion.** The sample was slowly added to the final concentration of 20% (m/v) trichloroacetic acid to precipitate protein, then vortexed to mix and incubated for 2 h at 4°C. The precipitate was collected by centrifugation at 4500 g for 5 min at 4°C. The precipitated protein was washed with pre-cooled acetone for 3 times and dried for 1 min. The protein sample was then redissolved in 200 mM triethyl-ammonium bicarbonate buffer (TEAB) and ultrasonically dispersed. Trypsin was added at 1:50 trypsin-to-protein mass ratio for the first digestion overnight. The sample was reduced with 5 mM dithiothreitol for 60 min at 37°C and alkylated with 11 mM iodoacetamide for 45 min at room temperature in darkness. Finally, the peptides were desalted by Strata X SPE column.

The quantification method used is LFQ (Label-Free Quantification), a non-labeling quantitative approach. Unlike some quantification methods that require stable isotopes or other labeling reagents, LFQ directly quantifies proteins based on mass spectrometry data. It determines the relative abundance of proteins by comparing the intensity of mass spectrometry signals (such as peak area or peak height) across different samples.

**LC-MS/MS analysis and database search for proteome.** The tryptic peptides were dissolved in solvent A (0.1% formic acid, 2% acetonitrile/ in water), directly loaded onto a home-made reversed-phase analytical column (25-cm length, 100 μm i.d.). Peptides were separated with a gradient from 6% to 24% solvent B (0.1% formic acid in acetonitrile) over 70 min, 24% to 35% in 14 min and climbing to 80% in 3 min then holding at 80% for the last 3 min, all at a constant flow rate of 450 nL/min on a nanoElute UHPLC system (Bruker Daltonics). The peptides were subjected to Capillary source followed by the timsTOF Pro (Bruker Daltonics) mass spectrometry. The electrospray voltage applied was 1.75 kV.

Precursors and fragments were analyzed at the TOF detector, with a MS/MS scan range from 100 to 1700 m/z. The timsTOF Pro was operated in parallel accumulation serial fragmentation (PASEF) mode. Precursors with charge states 0–5 were selected for fragmentation, and 10 PASEF-MS/MS scans were acquired per cycle. The dynamic exclusion was set to 30 s. The resulting MS/MS data were processed using MaxQuant search engine (v.1.6.15.0). Tandem mass spectra were searched against the Mus_musculus database (17063 entries) concatenated with reverse decoy database. Trypsin/P was specified as cleavage enzyme allowing up to 2 missing cleavages. The mass tolerance for precursor ions was set as 20 ppm in First search and 20 ppm in Main search, and the mass tolerance for fragment ions was set as 20 ppm. Carbamidomethyl on Cys was specified as fixed modification, and acetylation on protein N-terminal and oxidation on Met were specified as variable modifications. FDR was adjusted to < 1%.

**LC-MS/MS analysis and database search for acetylome.** The resulting peptide was dissolved in immunoprecipitation (IP) buffer (containing 100 mM NaCl, 1 mM EDTA, 50 mM Tris-HCl, 0.5% NP-40, pH 8.0). The supernatant was transferred to the pre-washed anti-acetylated Ab resin (No.3685542326611196740, PTM Biolabs Inc, Hangzhou, China), and incubated on a rotating shaking table at 4°C overnight. After incubation, the resin was washed four times with IP buffer solution, and then twice with deionized water. Subsequently, 0.1% trifluoroacetic acid was used to elute the resin-bound peptide via washing three times. The eluate was collected and vacuum freeze-dried. Finally, the samples were desalted (C18, Z720062, Millipore) and vacuum freeze-dried for liquid chromatography-mass spectrometry analysis.

The tryptic peptides were dissolved in solvent A, directly loaded onto a home-made reversed-phase analytical column (25-cm length, 100 μm i.d.). The mobile phase consisted of solvent A (0.1% formic acid, 2% acetonitrile/in water) and solvent B (0.1% formic acid in acetonitrile). Peptides were separated with following gradient: 0–42 min, 7%−24%B; 42–55 min, 24%−32%B; 54–57 min, 32%−80%B; 57–60 min, 80%B, and all at a constant flow rate of 450 nl/min on a NanoElute UHPLC system (Bruker Daltonics). The peptides were subjected to capillary source followed by the timsTOF Pro mass spectrometry. The electrospray voltage applied was 1.75 kV. Precursors and fragments were analyzed at the TOF detector, with a MS/MS scan range from 100–1700. The timsTOF Pro was operated in parallel accumulation serial fragmentation (PASEF) mode. Precursors with charge states 0–5 were selected for fragmentation, and 10PASEF-MS/MS scans were acquired per cycle. The dynamic exclusion was set to 30 s.

The resulting MS/MS data were processed using MaxQuant search engine (v.1.6.15.0). Tandem mass spectra were searched against Mus_musculus database (17063 entries) concatenated with reverse decoy database. Trypsin/P was specified as cleavage enzyme allowing up to 4 missing cleavages. The mass tolerance for precursor ions was set as 20 ppm in First search and 20 ppm in Main search, and the mass tolerance for fragment ions was set as 20 ppm. Carbamidomethyl on Cys was specified as fixed modification, and acetylation on protein N-terminal and oxidation on Met and acetytation on Lys were specified as variable modifications. False discovery rate (FDR) of protein and PSM was adjusted to < 1%.

## Bioinformatics analysis

Bioinformatics analysis was conducted based on the raw files from mass spectrometry, involving 1) construction of a specific protein database and database searching using analysis software; 2) quality control of the peptides and modification sites; 3) annotation of identified proteins using functional databases such as GO, KEGG, InterPro, COG, and STRING; 4) quantitative analysis of the modified sites, including quantitative distribution and repeatability analysis; 5) statistical difference analysis based on the difference screening of quantitative results; 6) motif analysis of the modified sites according to the differential analysis; 7) classification and statistically analysis of the differentially modified proteins, including GO secondary classification, KEGG pathway classification, and COG/KOG classification and statistics; 8) enrichment analysis using Fisher's exact test based on the statistical results using different classification methods; 9) enrichment cluster analysis of different groups to illustrate the functional relationship of different modified proteins; 10) protein-protein interaction (PPI) analysis to identify the key regulatory modification proteins. Differential acetylated proteins with fold change

> 1.5 or < 0.667 were assessed using the STRING (v.11.0) protein network database to extract interaction relationships, with a confidence score >0.7 (high confidence). Visualization was achieved through the R package "networkD3".

## Immunofluorescence staining

Immunofluorescence staining was performed following the manufacturer's instruction using a four-color multiple fluorescence staining kit (abs50028, Absin). Briefly, rat kidneys were fixed with 4% paraformaldehyde for 24 h at 4°C and then embedded in paraffin. The paraffin-embedded tissue was sectioned into 5-µm sections. After dewaxing and rehydrating, the sections were subjected to antigen retrieval by immersing them in 10 mM sodium citrate (pH 6.0) and microwaving at 1000 W for 30 min, followed by cooling to room temperature. The sections were blocked in 10% BSA at 37°C for 30 min. The primary antibody, appropriately diluted (the detailed information of the antibodies used in the study was listed in Table 1), was added to the slides and incubated at 4°C overnight. The following day, after three washes with tris-buffered saline, the sections were incubated with horse radish peroxidase-conjugated secondary antibody for 10 min at room temperature. After another three washes, 100 µL of signal-enhanced solution was added to the section and incubated at room temperature for 10 min. The slides were washed thrice with tris-buffered saline, and subjected to antigen retrieval again, followed by the second antibody incubation. Once all antibody incubations were completed, the sections were counterstained with Hoechst 33342 (1:2000; Invitrogen) for 5 min at room temperature. Following two additional washes, the sections were mounted in SlowFade (Invitrogen) and observed using a fluorescent microscope (magnification, 200X; Zeiss LSM 880 GmbH).

## Hematoxylin-eosin (HE) and von Kossa's staining

The kidneys were fixed with 4% paraformaldehyde, embedded in paraffin, and sectioned at 3-µm intervals, then dewaxed with xylene and dehydrated in ethanol and stained with HE (C0105, Beyotime) or Von Kossa (G3282, Solarbio) following the manufacturer's instructions. For HE staining, the slides were stained with hematoxylin for 10 min, and then wash with tap water for 10 min to remove residual staining solution. Hydrochloric ethanol was added to treat the slide for up to 5 seconds. After 10 min wash with tap water, the sections were stained with eosin for 1 min, and then subjected to another 10 min wash. For Von Kossa staining, slides were soaked in a 5% silver nitrate solution, exposed to ultraviolet light for 1 h, and rinsed with distilled water for 1 min. The sections were then placed in a 5% solution of sodium thiosulfate for 2 min, rinsed with distilled water, and stained with the nuclear fast red solution for 3 min. All HE and Von Kossa's staining slides were dehydrated with graded alcohols and xylene, and sealed with neutral resin. These slides were scanned with a Leica DMi8 Microsystem CMS GmbH (magnification, 20X).

## Western blotting

Twenty micrograms protein samples (determined by bicinchoninic acid) were loaded and run on 4–20% SDS-PAGE, and transferred onto a polyvinylidene fluoride (PVDF) membrane. The membrane was blocked with 5% (w/v) non-fat milk in

**Table 1. Antibody information.**

| No | Antibody | Company | CatLog | Dilutions |
|----|----------|---------|--------|-----------|
| 1 | Anti-CALM1 Rabbit Polyclonal Antibody | Absin | abs135839 | 1:200 |
| 2 | Anti-GSTK1 Rabbit Polyclonal Antibody | Proteintech | 14535-1-AP | 1:200 |
| 3 | Anti-ALBUMIN Mouse Monoclonal Antibody | Proteintech | 66051-1-Ig | 1:200 |
| 4 | Anti-NDUFS1 Mouse Monoclonal Antibody | Santa Cruz | sc-271510 | 1:200 |
| 5 | Anti-VINCULIN Rabbit Polyclonal Antibody | Servicebio | GB111328 | 1:200 |
| 6 | Anti-VPS4B Mouse Monoclonal Antibody | Santa Cruz | sc-377162 | 1:200 |
| 7 | Anti-Acetyl lysine Rabbit mAb | PTM Biolabs | PTM-105RM | 1:200 |

PBS buffer at room temperature for 0.5 h. After incubation overnight at 4°C with anti-acetylated lysine rabbit mAb (1:1,000 dilution), the membrane was treated with horseradish peroxidase-labeled secondary antibody (anti-rabbit IgG, 1:1000, cat nos. 7074, Cell Signaling Technology, Inc.) for 1 h at room temperature. Positive bands were detected using an enhanced chemiluminescence kit (Thermo Fisher Scientific, Inc.).

## Statistical analysis

Statistical analyses were performed and data are expressed as mean±standard error of the mean (SEM). The statistical significance between groups was determined using either one-way analysis of variance (ANOVA) followed by Bonferroni's post hoc test or unpaired Student's t-test, where a P-value of less than 0.05 was considered statistically significant. All statistical comparisons were made relative to the control group, with significant differences indicated in the corresponding figures.

## Results

### Model development and sample qualification

The urolithiasis model was successfully established in SD rats through ethylene glycol and ammonium chloride induction, followed by LFQ-based quantitative protein acetylation analysis (Fig 1A). Prior to analysis, HE and von Kossa's staining were conducted to detect CaOx deposits and tubule-interstitial damages in the rat kidneys. CaOx crystal deposits were indicated using red stars (Fig 1B), confirming the successful development of the CaOx calculi rat model. Coomassie Brilliant Blue Staining revealed no observable difference (Fig 1C), while the use of anti-acteyl lysine antibodies demonstrated differential expression bands between control and kidney calculi rats (Fig 1C). The detailed information of bands of interest using red boxes was included in S1 File in S1 Data. For protein identification, 260,225 total spectra were obtained, revealing 6,838 sites, of which 4,015 were quantifiable sites with corresponding to 1,565 quantifiable proteins (Fig 1D).

### Protein acetylation and differential proteomics profiling

Principal component analysis (PCA) revealed better quantitative repeatability among biological replicates than between different groups (Fig 2A and 2C). Triplicates of control and stones were clustered, and substantial differences were shown between the two groups. PC1 and PC2 accounted for 64.5% (Fig 2A), and 55.0% (Fig 2C) of the differences between the control and stone groups in the proteomics and acetylome, respectively. It was expected that there was a greater difference between the stone samples. For differential proteomics, 197,216 of 1,144,282 total spectrums was matched, peptides and unique peptides were 38,757 and 36756, respectively. 5065 proteins were identified, and the number of quantifiable proteins is 4002, of which 129 proteins were upregulated proteins and 118 proteins downregulated (Fig 2B). Among the 4,015 quantifiable sites corresponding to 1,565 quantifiable proteins, a total of 671 differential acetylated sites were identified in the kidney of calculi rat, with 133 upregulated and 538 downregulated sites corresponding to 118 and 258 proteins, respectively (Fig 2D). Significant differences were observed in proteomics and protein acetylation profiles between the control and stone groups. All the differential expressed proteins and acetylated sites and their proteins could be found in the S2 File in S1 Data.

Further analysis of the acetylome data revealed a volcano plot showcasing differentially modified proteins with fold change > 2 or < 0.5 (Fig 3A). Notably, proteins with more than five acetylated sites, such as Hspa9, Clybl, Ldh2, Aldh6a1, Fh, Mah2, Acat1, Glud1, Aco2, Slc25a5, Got2, and Shmt2, were prominently displayed. Fig 3B revealed modified proteins with both upregulated and downregulated sites. The distribution of modification sites per protein is illustrated in Fig 3C. A heatmap showed 20 upregulated acetylated sites in the kidney calculi samples (Fig 3D).

### Functional enrichment analysis of differentially expressed and modified proteins

GO category analysis was conducted to evaluate the critical terms of the differentially expressed (Fig 4) and modified proteins (Fig 5) related to kidney stone formation. The detailed information refers to the proteins and sites were provided

**Fig 1. Histochemical validation of the calcium crystals in the calculi rat model.** A, Flow chart of the study. B, HE staining and von Kossa's staining of rat model kidney. Original magnification, ×40. C, Coomassie Brilliant Blue Staining and Western blotting using an anti-acetyllysine antibody (1: 1000 dilution, PTM-105RM, PTM Biolabs). Equal amount proteins were from control and kidney calculi rats (n = 3). The bands of interest were highlighted

using red boxes. D, An overview of LFQ-based quantitative protein acetylation identification. 260,225 total spectra were identified, revealing 6,838 sites, of which 4,015 were quantifiable sites with corresponding to 1,565 quantifiable proteins.

in the S3 File in S1 Data. In Fig 4A, 206, 157, and 156 differentially expressed proteins were involved in cellular process, biological regulation, and metabolic process, respectively. Binding (161 proteins) and catalytic activity (105 proteins) were the dominant molecular functions. Fig 4B highlighted more upregulated proteins in the cytoskeleton (13 upregulated and 1 downregulated), signal transduction mechanisms (30 upregulated and 4 downregulated), posttranslational modification, protein turnover, chaperones (14 upregulated and 5 downregulated), while more downregulated proteins were evident in translation, ribosomal structure and biogenesis (23 downregulated and 1 upregulated), amino acid transport and metabolism (12 downregulated and 5 upregulated), and energy production and conversion (11 downregulated and 5 upregulated). The list of the all the proteins was showed in the S3 File in S1 Data.

In Fig 5A, similar to the proteomics data, most proteins were associated with cellular processes (320 proteins), metabolic processes (281 proteins), and biological regulation (189 proteins) in terms of biological processes. Regarding the molecular functions, the majority were enriched in catalytic activity (234 proteins) and binding (229 proteins). Notably, proteins (19 proteins) related to reproduction were downregulated. Clusters of Orthologous Groups of protein analysis (Fig 5B) highlighted proteins involved in energy production and conversion (84 proteins, 78 upregulated and 10 downregulated), posttranslational modification, protein turnover, chaperones (43 proteins, 20 upregulated and 24 downregulated), and lipid transport and metabolism (39 proteins, 37 upregulated and 2 downregulated). Subcellular localization analysis (Fig 5C) predominantly identified protein in the mitochondria (152 proteins) and cytoplasm (121 proteins), with mitochondrial proteins primarily upregulated (143 proteins were upregulated and 12 proteins were downregulated) in the renal calculi rats compared to controls, along with a minor downregulation in the endoplasmic reticulum (4 sites). The list of the all the proteins was showed in the S3 File in S1 Data.

## PPI network analysis showed the interaction of differentially expressed and modified proteins

PPI network analysis of differentially expressed and modified proteins was conducted as described before. S1 Fig in S1 Data displayed the interaction of differentially expressed proteins in the Top 5 pathways. S2 Fig in S1 Data presents the relationship of the differentially expressed proteins with the pathway. S3 and S4 Figs in S1 Data show the interaction of the differentially modified proteins in the Top 5 pathway and the relationship of the differentially modified proteins with the pathway, respectively.

## Differentially expressed and acetylated proteins were predominantly linked to metabolic process

Most of the differentially expressed and acetylated proteins were involved in cellular and metabolic processes. The GO analysis highlighted the Top 20 pathways associated with metabolic processes (Fig 6A). The subcellular localization of the proteins in the pathways was displayed in Fig 6C, with cytoplasm (29.13%) and mitochondria (55.34%) being dominant subcellular fractions. The Top 20 KEGG pathways with the keyword metabolism (Fig 6B), cytoplasm, and mitochondria proteins accounted for 29.13% and 61.17% of the total proteins, respectively (Fig 6D). Further investigation into mitochondrial differential modified proteins emphasized pathways predominantly linked to metabolic processes (Fig 7A and 7B).

## Immunofluorescence confirms the differentially modified proteins from the acetylome data

The mass spectrometry data were verified in a previous study [16], and the acetylation levels of the differentially modified proteins were further validated through immunofluorescence. As shown in Fig 3C, protein with one modification site was the dominate. According to the differential modified data in S1 File in S1 Data, we selected three up-regulated and three

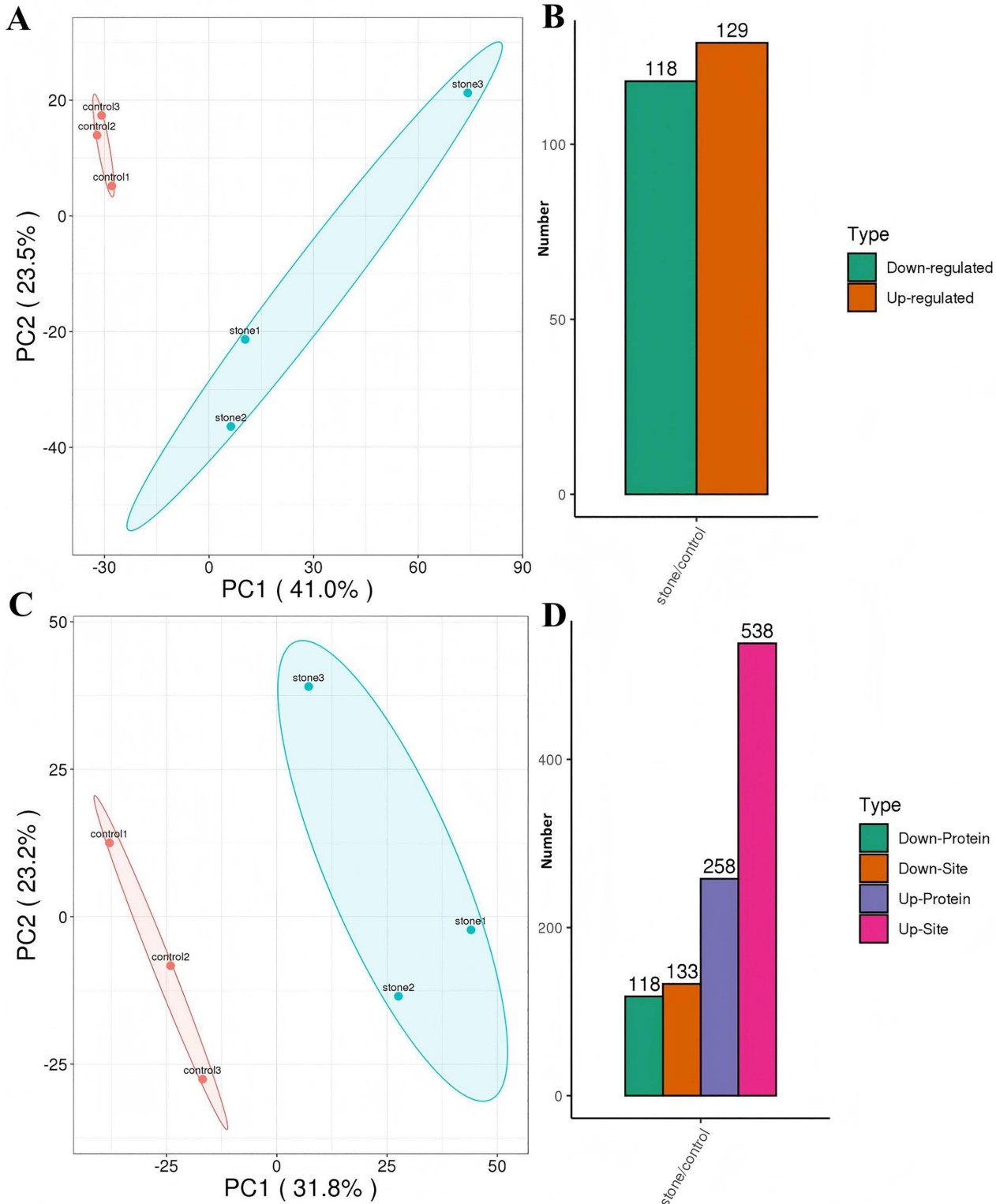

**Fig 2. Cluster analysis of differentially expressed proteins and modified sites in the kidney of calculi rat.** A & C, Principal component analysis (PCA) to evaluate the repeatability of protein quantification, A for proteomics (n = 3), and C for acetylome (n = 3). B & D, An overview of differential expressed proteins and sites in the kidney of calculi rat. B for proteomics (n = 3), and D for acetylome (n = 3).

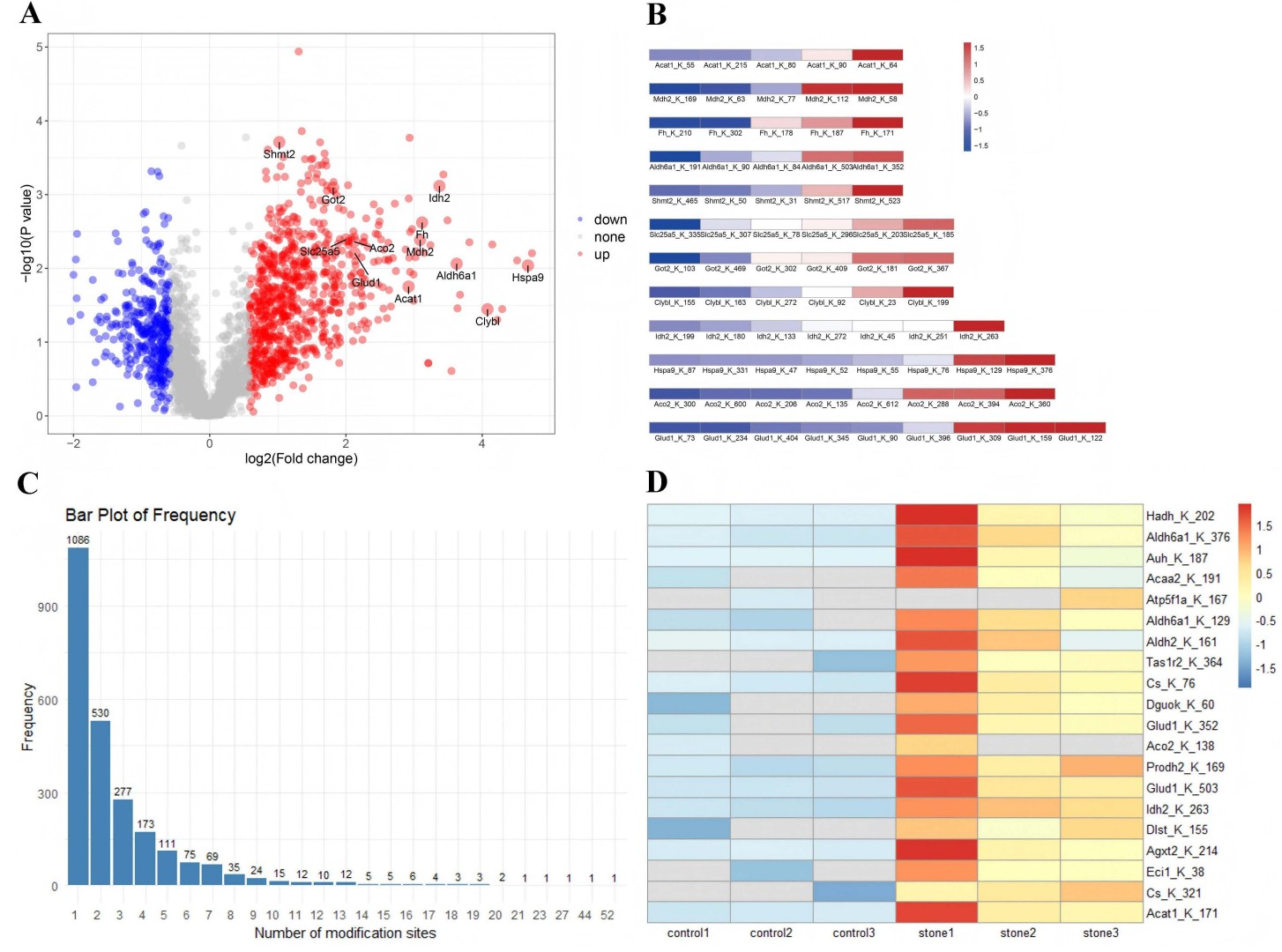

**Fig 3. An overview of acetylome data.** A, Volcano plot showcasing differentially modified proteins with fold change > 2 or < 0.5. B, Represented modified proteins with both of upregulated site and downregulated site. C, The number of modification sites per proteins. D, Heatmap of 20 upregulated acetylated sites in the kidney calculi samples.

down-regulated acetylated proteins with one modification site randomly, whose protein expresseion level was unchanged (Proteomics data in S1 File in S1 Data) for verification. As shown in Fig 8, consistent with the acetylome data (Fig 8A), proteins with only one acetylated sites like CALM1 (Kac22), GSTK1 (Kac165), and ALBUMIN (Kac588) were upregulated in the kidney calculi rats. Consversely, NDUFS1 (Kac607), VINCULIN (Kac666), and VPS4B (Kac217F) were down regulated in the kidney calculi rats (Fig 8B).

## Discussion

Nephrolithiasis is an increasingly prevalent systemic disorder with significant health and economic consequences [17]. Among kidney stones, CaOx stones prevail as the most common type, posing a substantial risk factor for chronic kidney

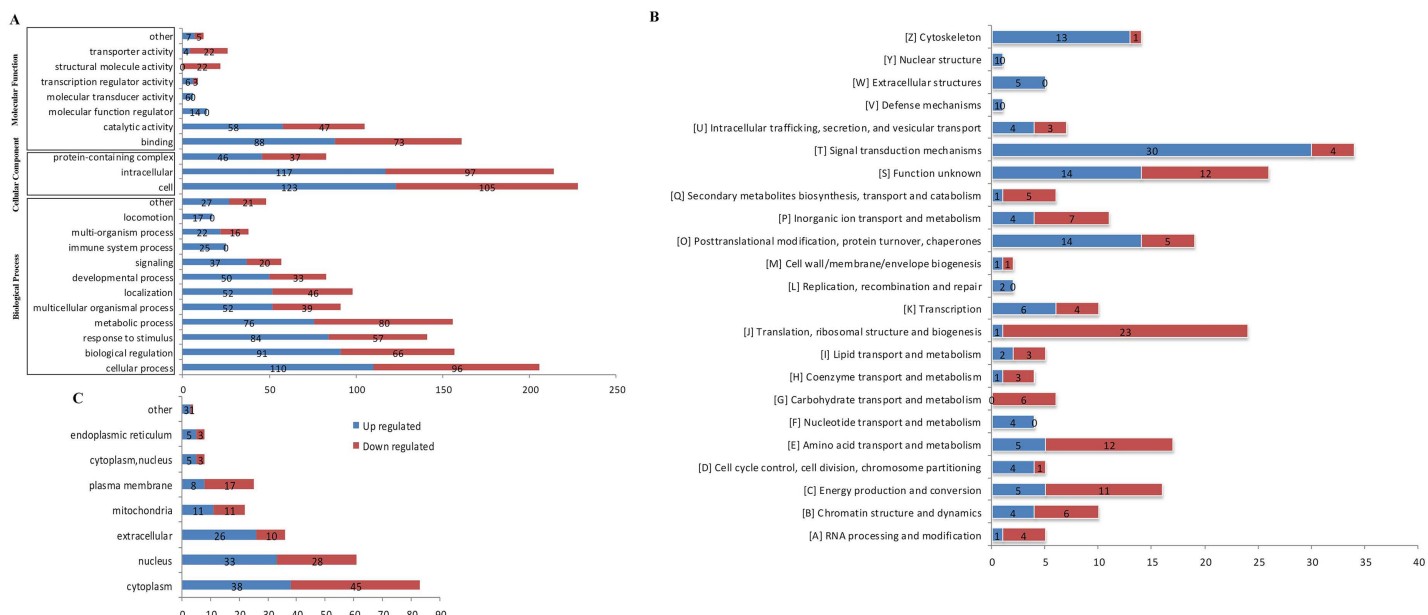

**Fig 4. Functional classification of differentially expressed proteins.** A, GO category analysis. The numbers of differentially expressed proteins involved in cellular process, biological regulation, and metabolic process, respectively. B, Clusters of Orthologous Groups of proteins analysis. C, Sub-cellular localization of the differentially expressed proteins.

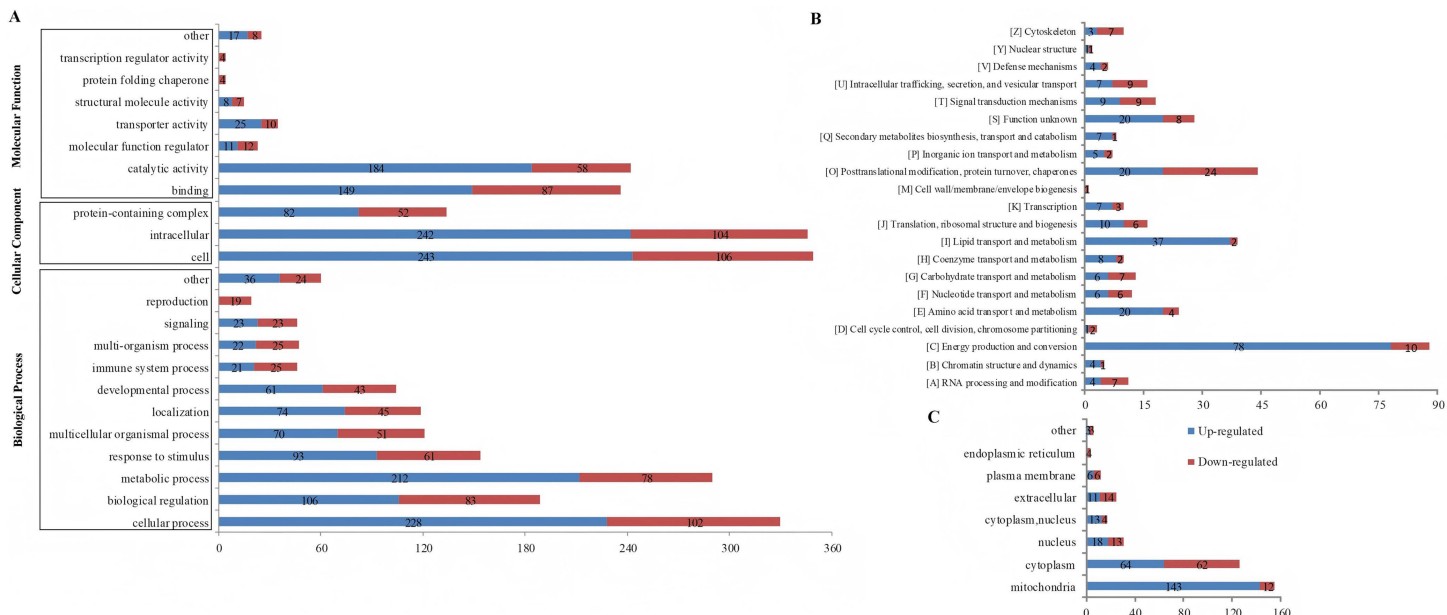

**Fig 5. Functional classification of differentially modified proteins.** A, GO category analysis. The numbers of differentially modified proteins involved in cellular process, biological regulation, and metabolic process, respectively.B, Clusters of Orthologous Groups of proteins analysis. C, Subcellular localization of the differentially modified proteins.

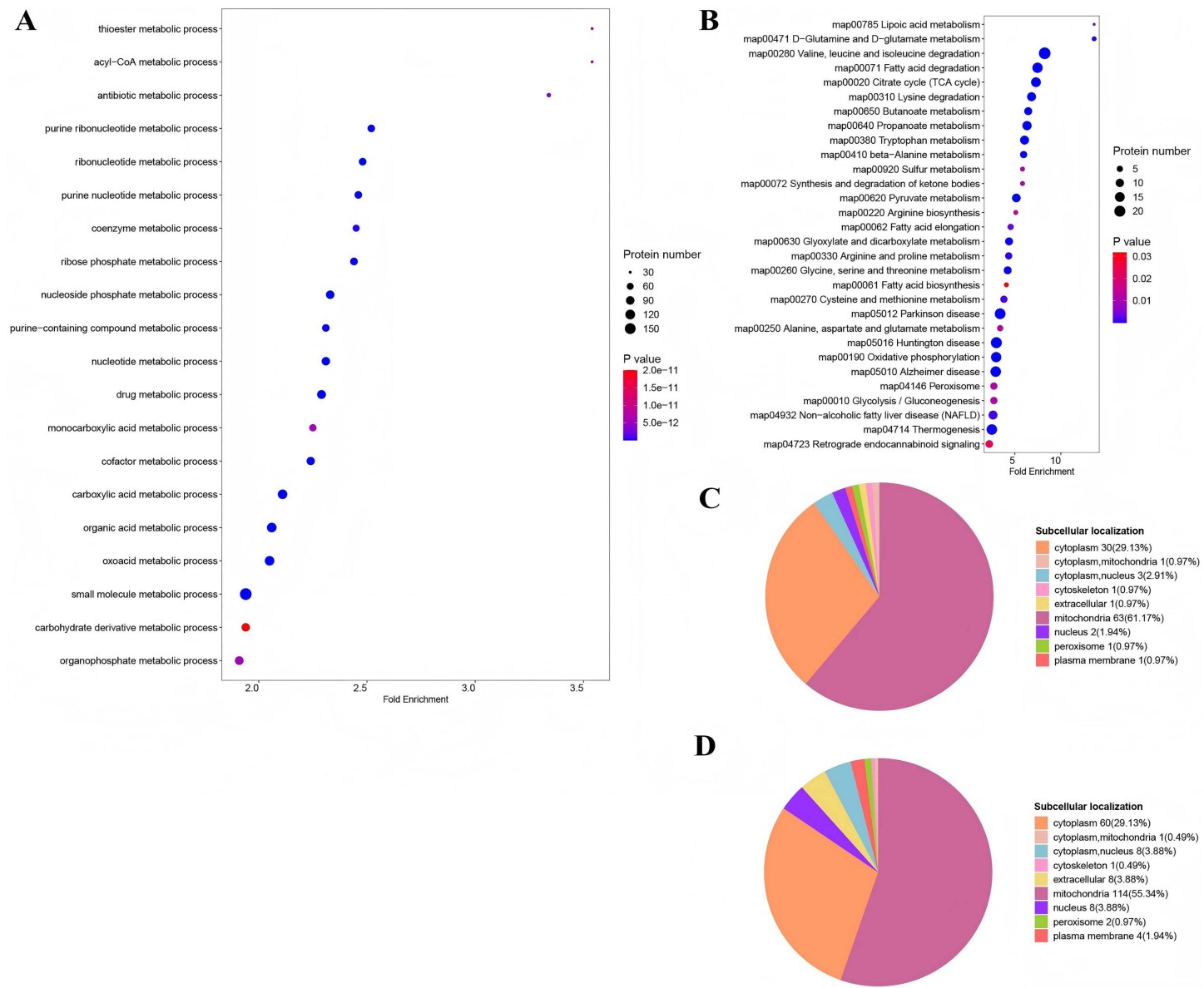

**Fig 6. Proteins in TOP GO and KEGG pathways referred to metabolism were localized in mitochondria.** Most of the differentially expressed and acetylated proteins were involved in cellular and metabolic processes. The top 20 GO (A) and KEGG (B) pathways refer to metabolism. Subcellular localization of the proteins in the top 20 GO (B) and KEGG pathways (D) refer to metabolism. Cytoplasm and mitochondria being the dominant subcellular fractions.

disease (CKD) and renal fibrosis [18,19]. Despite significant advancements in understanding its pathophysiology, existing therapeutic options possess limitations.

Numerous *in vitro* and *in vivo* models were developed that attempt to replicate human urolithiasis. Transgenic mouse with selective knockout of osteopontin (OPN) [20], Tamm–Horsfall protein [21], and cysteine transporter [22], as well as in *vivo Drosophilia* genetic model [23] of CaOx crystals were generated, however, the overall accuracy and consistency in relation to human kidney stone disease remain controversial [24]. Rat model of urolithiasis represents a well-established,

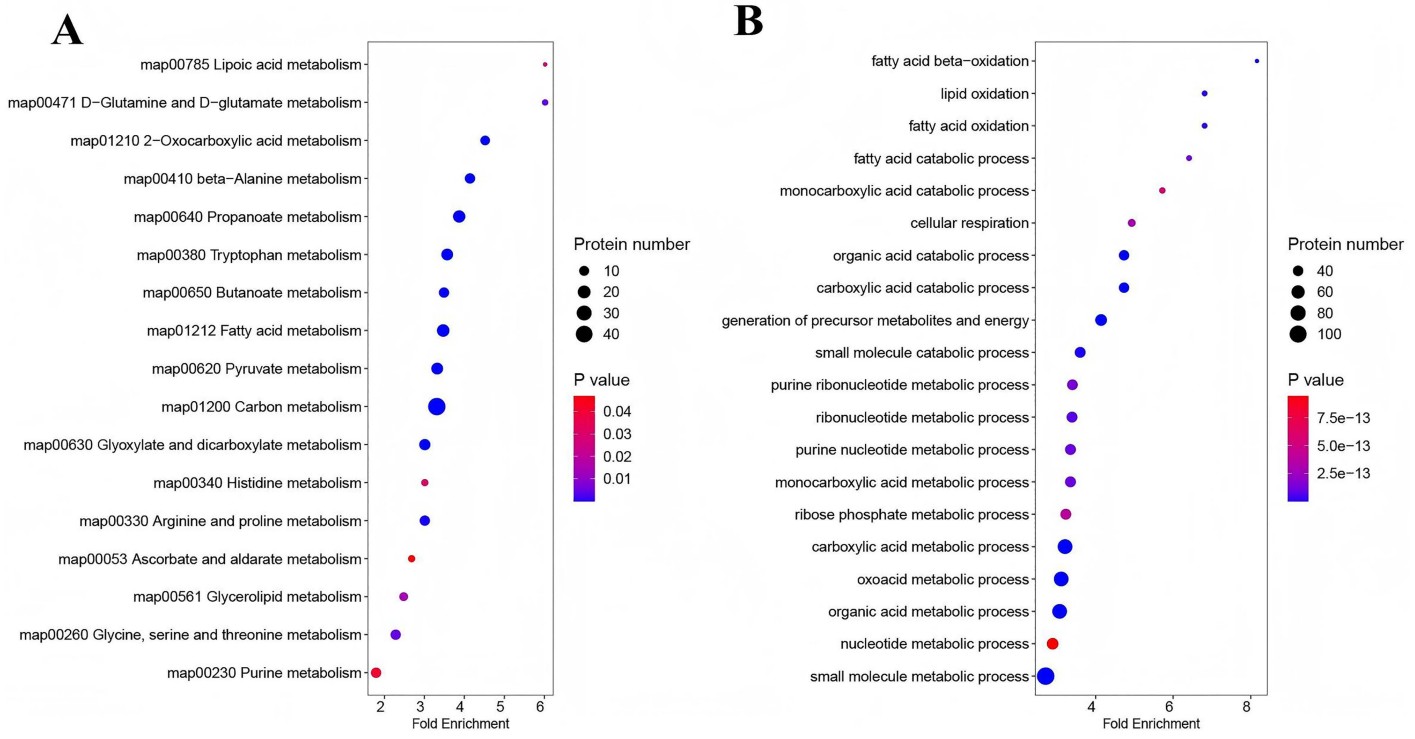

**Fig 7. Functional classification of mitochondria differential modified proteins.** A, KEGG pathways. B, Top 20 GO pathways. Both pathways predominantly linked to metabolic processes.

relatively economic model [24]. Ethylene glycol is one of the most common models for inducing urinary crystals in rats [25]. Multiple rodent studies have shown that testosterone amplifies crystal deposition by up-regulating glycolate oxidase and increasing urinary oxalate excretion, whereas estrogen is protective. Consequently, male rats develop larger and more consistent CaOx burdens within the same experimental timeframe, reducing inter-individual variability and the sample size required for proof-of-concept studies.

Network and GO analyses have emerged as robust tools for comprehending complex diseases [26,27], providing a visual framework and protein enrichment for specific functional categories [28]. Many proteins have been reported to be involved in kidney stone formation, such as OPN [29], matrix Gla protein (MGP) [30], bikunin [31], and Tamm-Horsfall proteins [21], detected in urine and kidney stone matrixes. Variants in these protein's genes have been linked to kidney stone disease risk [32–34]. Recently, more than 1000 proteins were identified using a urinary proteome analysis [35], and these proteins interacted with each other to play a vital role in modulating crystal nucleation, growth, aggregation, and adhesion to renal epithelial cells.

Omics studies of kidney stones exhibit substantial advantages in higher throughput and larger scale analysis compared to traditional studies. Posttranslational modification refers to the covalent modification of proteins following protein translation, and acetylation is one of the most commonly studied types. Over 2000 lysine-acetylated human proteins have been identified [36], and protein acetylation is implicated in a multitude of functions at all levels of biological processes [37]. To the best of our knowledge, the present study is the first to analyze kidney tissue acetylome spectrum, revealing hyper-acetylated kidney proteins in kidney calculi rats. We identified 671 differential acetylated sites, with 538 upregulated sites corresponding to 258 proteins and 133 downregulated sites corresponding to 118 proteins.

Increasing evidence has illustrated the role of acetylation in kidney injury pathologies. Increased acetylation levels of p65 (NF-κB) and STAT3 have been detected in mouse and human diabetic kidneys [38], while p53 deacetylation has been

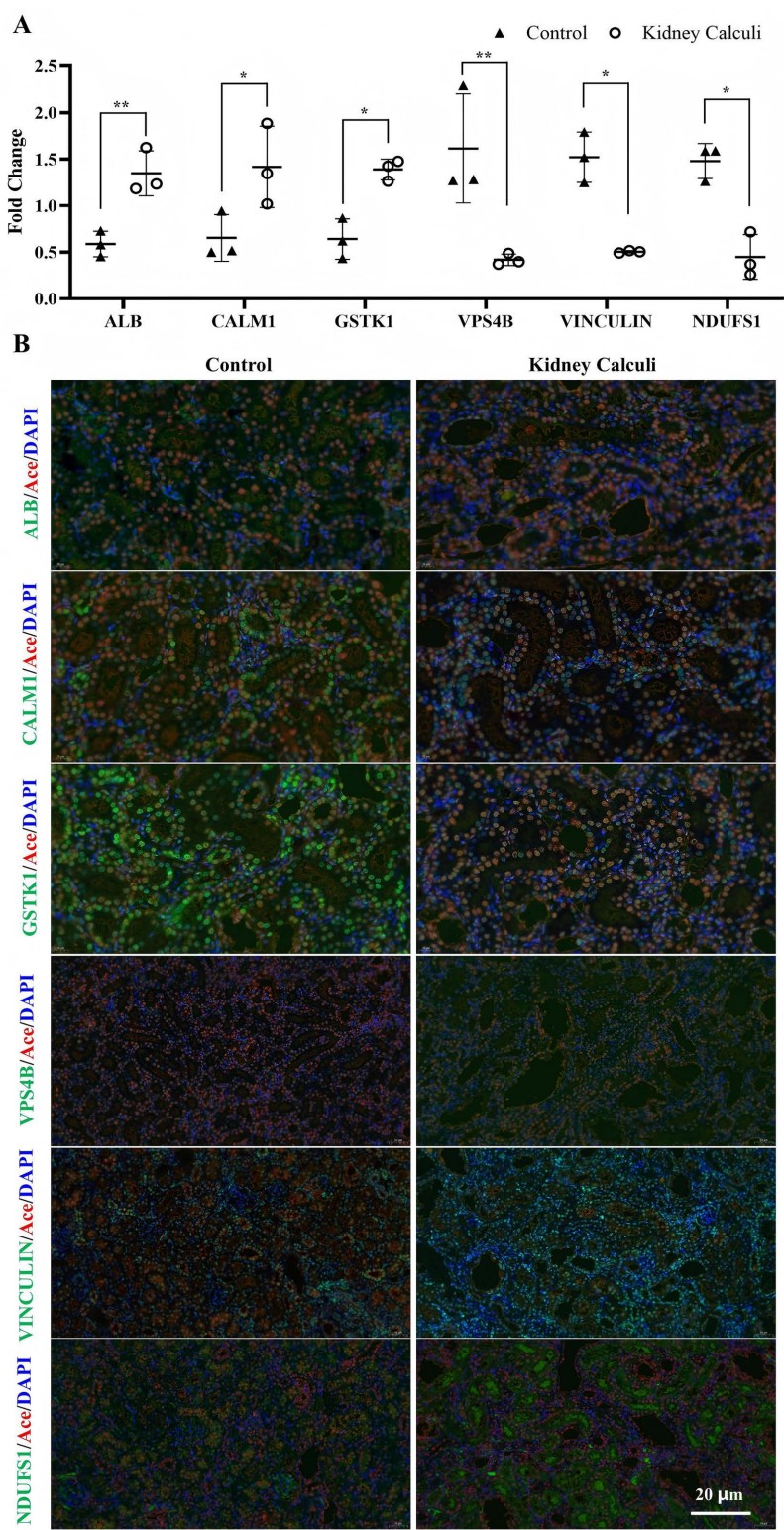

**Fig 8. Immunofluorescence of differential acetylated proteins.** The acetylation levels of the differentially modified proteins (A) dectected by LC-MS were validated through immunofluorescence **(B)**. ALBUMIN (Kac588). CALM1 (Kac22). GSTK1 (Kac165). VPS4B (Kac217). VINCULIN (Kac666). NDUFS1 (Kac607). Red, Acetylysine. Blue, Hoechst 33342 for nuclei. Green, target protein. Magnification 200X. *, P < 0.05; **, P < 0.01.

shown to alleviate calcium oxalate deposition-induced renal fibrosis by inhibiting ferroptosis [39]. Moreover, acetylated p53 expression levels were seen to correlate with changes in SLC7A11 (increased) and GPX4 (decreased) under oxalate stimulation [39], suggesting a promotional effect of acetylation in stone formation. Furthermore, increasing H3K9 and H3K27 acetylation has been reported to promote the expression of miRNAs (microRNAs-130a-3p, -148b-3p, and -374b-5p), enhance urinary citrate excretion, and reduce urinary calcium excretion by suppressing Nadc1 and Cldn14 expression [8]. Consistently, this study also reported an inhibitory effect of acetylation in stone formation.

Metabolic process emerged as the most prevalent biological process, with acetylation being a significant posttranslational modification influencing mitochondrial metabolism. Most differentially expressed proteins were localized in the mitochondria, with 143 upregulated and 12 downregulated proteins. Honestly, it was not surprising that the subcellular localization of the differential expressed and modified proteins was dominated in the mitochondria. Mitochondrial dysfunction is one of the main causes of cell apoptosis caused by calcium oxalate stones [40–42]. When come to the compare with the control, the mitochondria proteins were upregulated (26 protein/11 proteins) and hyper-acetylated (365 sites/13 sites) in the stone groups. The increased acetylation of mitochondrial proteins (55.43%) was found to influence various mitochondrial pathways, leading to significant changes in mitochondrial morphology and collagen deposition in kidney tissues of nephrolithiasis and hyperoxaluric mice [39].

However, our study does have certain limitations. Ethylene glycol is the precursor of oxalate, ammonium chloride could acidify urine, and lead to renal tubular dysfunction, thus the animal model induced by ethylene glycol and ammonium chloride may not fully reflect the patient's true condition. The clinical application of the protein acetylation using human derived cells (kidney organoids) and patient samples are urgent needed, and the proteome and acetylome data provided numerous candidate targets for studying and clarifying the mechanism of the occurrence and development of kidney stones. Additionally, having only three repeated samples in each group could lead to biased results, but the good reproducibility between samples can also support the findings of the study to some extent. Given the pilot nature of this study (N = 3 per group), these findings should be regarded as preliminary; a larger, adequately powered study is warranted before drawing definitive conclusions. More, importantly, the molecular mechanisms of acetylated proteins in kidney calculi formation require further research. Due to the limitation of this study and extend these findings beyond the animal model, we are currently obtaining de-identified serum and tissue specimens from patients with renal stones, and planning to verify the data using patients' samples. Integrating these human samples will allow us to corroborate the mechanistic pathways identified here and will provide more evidence to support the involvement of acetylation modification in the nephrolithiasis, and the acetylation modification sites identified in the present study may be selected as critical drug therapeutic targets of renal stones.

In conclusion, our LC-MS/MS investigation of protein expression profiles and lysine acetylation features in kidney tissues from kidney calculi and normal control rats, has provided insights into the landscape of acetylated proteins involved in the formation of renal stones, and a data resource for better understanding of the pathogenesis of nephrolithiasis. We characterized the calculi oxalate crystals related protein lysine acetylation expression profiles in the kidney. Differential expressed and modified proteins identified in the study will contribute to further clarifying the molecular mechanism underpinning renal stone formation and development. As the first comparative study of Kac modification associated with kidney calculi, the integrated proteomic and network analyses identify mitochondrial energy failure as an early and specific signature of calcium-oxalate nephropathy. This finding pinpoints the electron-transport chain as a tractable target for future mechanistic studies and, ultimately, therapeutic intervention in patients with kidney stone disease.

## Supporting information

**S1 Data.** S1 Fig. Interaction of differentially expressed proteins in the Top 5 pathway. A, biological process. B, cellular component. C, KEGG. D, molecular function. S2 Fig. The relationship of the differentially expressed proteins with the pathway. A, biological process. B, cellular component. C, KEGG. D, molecular function. S3 Fig. Interaction of differentially

modified proteins in the Top 5 pathway. A, biological process. B, KEGG. C, cellular component. D, molecular function. S4 Fig. The relationship of the differentially modified proteins with the pathway. A, biological process. B, KEGG. C, cellular component. D, molecular function. S1 File. The detailed information of bands of interest using red boxes in Fig 1C. S2 File.The differential expressed proteins and acetylated sites. S3 File. The detailed information refers to the proteins and sites in Figs 4 and 5.
(ZIP)

## Author contributions

**Conceptualization:** Zhu Wang, Qiong Deng.

**Data curation:** Meiyu Jin.

**Formal analysis:** Jieyan Wang.

**Investigation:** Shiwei Zhang, Zhu Wang, Qiong Deng.

**Methodology:** Zhu Wang.

**Software:** Meiyu Jin.

**Supervision:** Hao Jiang, Hui Liang, Qiong Deng.

**Validation:** Hao Jiang, Hui Liang.

**Visualization:** Jieyan Wang, Meiyu Jin.

**Writing – original draft:** Shiwei Zhang.

**Writing – review & editing:** Hui Liang, Qiong Deng.

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
