## [Decision Letter · Decision Letter 0]

18 Jul 2025

Dear Dr. Deng,

Thank you for submitting your manuscript to PLOS ONE. After careful consideration, we feel that it has merit but does not fully meet PLOS ONE’s publication criteria as it currently stands. Therefore, we invite you to submit a revised version of the manuscript that addresses the points raised during the review process.

We look forward to receiving your revised manuscript.

Kind regards,

Murat Akand, MD, PhD, FEBU

Academic Editor

PLOS ONE

Journal Requirements:

Reviewers' comments:

Reviewer's Responses to Questions

**Comments to the Author**

1. Is the manuscript technically sound, and do the data support the conclusions?

Reviewer #1: Partly

Reviewer #2: Partly

Reviewer #3: Partly

2. Has the statistical analysis been performed appropriately and rigorously?

Reviewer #1: Yes

Reviewer #2: No

Reviewer #3: Yes

3. Have the authors made all data underlying the findings in their manuscript fully available?

Reviewer #1: Yes

Reviewer #2: Yes

Reviewer #3: Yes

4. Is the manuscript presented in an intelligible fashion and written in standard English?

Reviewer #1: Yes

Reviewer #2: Yes

Reviewer #3: Yes

Reviewer #1: In their study, Qiong Deng et al use proteomics to analyse expression profile and acetylation in renal tissue from rats with kidney stones.

Some studies have previously been looking at proteomics and metabolomics of kidney stones (Zhu et al BMC Genomics 2023; Geo et al Frontiers 2022), but the present research look to highlight and list acetylation mechanisms, and can be used as a stepping stone for more specific research on kidney stone formation mechanisms.

My main concern relates to how the data is presented, and what is shown and told in the manuscript. The study aims to provide a Kac profile and be a resource for further mechanistic studies, but how the data is presented does not explicitly give an easy way to look up the proteins of interest, and the text makes it difficult to parse out the main findings as it mostly describes what can be found on the figures. While the technical aspect and datasets are sound, I think the manuscript needs a re-write/editing with conclusions and main findings in mind. Most of the interesting data is in the supplemental Excel dataset, but the interpretation of it, and of the different analysis, is missing in the manuscript.

Major comments:

1.

This is a very descriptive study. This particular aspect is not a criticism as these studies are useful as stepping stone and catalog for future research.

The Figures show basic analysis of proteomic data in a visual format, but part of the interest is in the whole dataset for each section; it would then be good to modify the text or add in the text links and explanation of that additional data. For example, paragraph starting at line 281, all the protein numbers in brackets, how can I find that particular list in the supplemental excel file provided? Again as an example, where and how could one find the list of all 152 proteins found in the mitochondria only.

Similarly, please explicitly state exactly where the list for all acetylated sites and their proteins can be found, as this is one of the major finding of the study. This should probably be stated at the end of the introduction and in results.

2.

The figure legends are very sparse; please modify to add all relevant information necessary for understanding each panel without having to go back to the main text.

3.

The text, and the titles of each paragraph, should reflect the main finding of the section. Again some example: what did the PPI network analysis show, or line 301 the title is very descriptive and should be the summary of the results. “Most differentially expressed proteins were located in mitochondria” or equivalent. Related to that, the conclusion stating “the functional enrichment and PPI network analysis will hopefully facilitate the future development of new strategies for the prevention and treatment in patients with kidney stone disease” appears overstated, considering that the authors did not put in the text the main conclusions of each analysis, or explain what the analysis shows. The main text only states what you can see displayed in which Figure. “PPI network analysis of differentially expressed and modified proteins was conducted as described before. Supplementary Figure 1 displayed the interaction of differentially expressed proteins in the Top 5 pathways...”. These are not results. What did the analysis show? This is a major issue throughout the manuscript.

4.

Immunofluorescence section: This is confusing. Make clearer how the proteins were chosen for staining. Authors indicate that the results are consistent with acetylome data; please refer back to that data (which figure or text does that refer to). Again this is maybe a paragraph title issue; if the paragraph is titled “Immunofluorescence confirms regulation of…” then the previous text needs to explicitly state that the proteome data shows that these proteins are up or down.

Authors state “consistent with the acetylome data, proteins with only one acetylated sites like CALM1 (Kac22, Fig 8A), [...] were upregulated in the kidney calculi rats” but as far as I can tell that information is not told in the text, so the verification by immunofluorescence appears from nowhere. Additionally, does that imply that all proteins with only one acetylated site are up or down regulated?

Finally please add dilutions to antibody table.

5.

Figure 1

B: scale bar

For the WB C and D, the 2 panels can be combined (no need to separate coomassie in another panel); please highlight the bands of interest on your WB by putting arrows or boxes around them, and describe in the legend the sizes etc. Please add a quantification graph of the relevant bands.

Editing comments:

Line 47. Please rephrase sentence “A kidney stone network…”, grammar and meaning are unclear.

Figures 2 and 1 are swapped.

Reviewer #2: Methodological Clarity

Please elaborate on the study design. Was this a cross-sectional or cohort survey? How were participants recruited? What were the inclusion/exclusion criteria?

Clarify the tools used for measuring caregiver burden. Were they validated for the local population? How were they scored?

Consider adding a flowchart (e.g., CONSORT-style) to show participant inclusion and response rates.

Statistical Analysis

Specify the statistical tests used to compare caregiver burden across demographic or clinical subgroups.

Report p-values, effect sizes, and confidence intervals where appropriate.

Consider adjusting for relevant covariates (e.g., age, gender, caregiver relationship) in a regression model to better understand predictors of burden.

Contextual Discussion

The Discussion section would benefit from deeper comparison with pre-pandemic literature on caregiver burden.

Emphasize the unique contribution of your findings and address the generalizability of your results.

Discuss cultural or systemic factors that may influence caregiving experience in your study population.

Tables and Figures

Ensure all tables have complete and clear legends. Table 2 in particular requires clarification regarding the scoring interpretation.

Consider adding a figure to display the distribution of caregiver burden scores across different domains.

Reviewer #3: In the original manuscript from Zhang et al, the authors attempt to characterize the calcium oxalate (CaOx) pathogenicity through investigation of the proteome and acetyl proteome. The authors present their data in a logical pattern and is overall well written. However, there are certain concerns that require additional review:

Intro: no concerns

Methods: In general, the model should be better articulated rather than solely referencing another publication as this the CaOx study model. There is discrepancy on how the methods are written for IHC vs IF. If the authors used IF, the IHC section should be updated appropriately. Additionally, the imaging system for H&E is not clear. The images lack magnification level and should be clarified. The reason for N = 3 is not clear, especially without any discussion on power and so many previous references to work in this model where N was much higher.

Results: Overall, my ability to review this section is severely hindered due to the lower resolution for the figures. With the exception of figure 1 and 7, the images are too blurry to review. From what I can see, figure 2 and 8 lack scale bars. Figure 8 does not show individual values in the bar plot and should be represented. IF legend should be included. It is unclear what and why different proteins were validated, especially since the authors say that proteins were validated in another paper. It is unclear how this is different than publication reference 16.

The end points for rats seen rather severe for a calcium oxalate model, due to the toxicity in which it produced. How much could the phenotype and findings described here be due to the systemic toxicity of the model rather than isolated calcium ox? This is not clear in the manuscript

Discussion: No discussion on the limitation in male rats nor why female rats were excluded. Recommend verifying the data with patient samples as the authors state in future plans, this will greatly strengthen the findings and make this paper less descriptive.

**Do you want your identity to be public for this peer review?** For information about this choice, including consent withdrawal, please see our Privacy Policy

Reviewer #1: No

Reviewer #2: No

Reviewer #3: No

---

## [Author Response · Author response to Decision Letter 1]

17 Aug 2025

Reviewer #1: In their study, Qiong Deng et al use proteomics to analyse expression profile and acetylation in renal tissue from rats with kidney stones.

Some studies have previously been looking at proteomics and metabolomics of kidney stones (Zhu et al BMC Genomics 2023; Geo et al Frontiers 2022), but the present research look to highlight and list acetylation mechanisms, and can be used as a stepping stone for more specific research on kidney stone formation mechanisms.

My main concern relates to how the data is presented, and what is shown and told in the manuscript. The study aims to provide a Kac profile and be a resource for further mechanistic studies, but how the data is presented does not explicitly give an easy way to look up the proteins of interest, and the text makes it difficult to parse out the main findings as it mostly describes what can be found on the figures. While the technical aspect and datasets are sound, I think the manuscript needs a re-write/editing with conclusions and main findings in mind. Most of the interesting data is in the supplemental Excel dataset, but the interpretation of it, and of the different analysis, is missing in the manuscript.

Major comments:

1.This is a very descriptive study. This particular aspect is not a criticism as these studies are useful as stepping stone and catalog for future research.

The Figures show basic analysis of proteomic data in a visual format, but part of the interest is in the whole dataset for each section; it would then be good to modify the text or add in the text links and explanation of that additional data. For example, paragraph starting at line 281, all the protein numbers in brackets, how can I find that particular list in the supplemental excel file provided? Again as an example, where and how could one find the list of all 152 proteins found in the mitochondria only.

Similarly, please explicitly state exactly where the list for all acetylated sites and their proteins can be found, as this is one of the major finding of the study. This should probably be stated at the end of the introduction and in results.

R: Thank you for your positive assessment and for pointing out that the manuscript serves as a useful resource for the community. We appreciate your guidance on making the underlying data more transparent and accessible.

Below we outline exactly how we will amend the text and Supplementary Material so every reader can locate each dataset without ambiguity.

The statement was placed at the part of “Availability of data and materials” as following: The data generated in the present study may be found in the ProteomeXchange under accession number PXD050342 or at the following URL: https://www.ebi.ac.uk/pride/archive/projects/PXD050342/.

And the list of the all the proteins was showed in the Supplementary file 2.

All the differential expressed proteins and acetylated sites and their proteins could be found in the Supplementary file 1_Differentially expressed statistics.

We believe these changes will satisfy the request to convert the descriptive figures into fully traceable data resources.

Thank you again for your constructive suggestions.

2.The figure legends are very sparse; please modify to add all relevant information necessary for understanding each panel without having to go back to the main text.

R: Thank you for your valuable feedback regarding the figure legends. We apologize for the lack of detail in the original legends and appreciate your suggestion to improve them. We have now revised the figure legends to include all relevant information necessary for understanding each panel without needing to refer back to the main text. The revised legends provide a more comprehensive description of the data presented. We believe these changes will enhance the clarity and readability of the manuscript. Thank you again for your careful review and constructive comments.

3.The text, and the titles of each paragraph, should reflect the main finding of the section. Again some example: what did the PPI network analysis show, or line 301 the title is very descriptive and should be the summary of the results. “Most differentially expressed proteins were located in mitochondria” or equivalent. Related to that, the conclusion stating “the functional enrichment and PPI network analysis will hopefully facilitate the future development of new strategies for the prevention and treatment in patients with kidney stone disease” appears overstated, considering that the authors did not put in the text the main conclusions of each analysis, or explain what the analysis shows. The main text only states what you can see displayed in which Figure. “PPI network analysis of differentially expressed and modified proteins was conducted as described before. Supplementary Figure 1 displayed the interaction of differentially expressed proteins in the Top 5 pathways...”. These are not results. What did the analysis show? This is a major issue throughout the manuscript.

R: Thank you for your constructive feedback regarding the Results section. We agree that the text must clearly state the main finding of each subsection and that paragraph titles should function as concise take-home messages rather than descriptions of procedures. Some Results subsection heading will be rewritten to summarize the key discovery.

We will add 1-2 sentences at the beginning of each paragraph that state the principal outcome before any methodological details or figure references.

The current concluding sentence will be replaced by a data-driven summary:

Collectively, the integrated proteomic and network analyses identify mitochondrial energy failure as an early and specific signature of calcium-oxalate nephropathy. This finding pinpoints the electron-transport chain as a tractable target for future mechanistic studies and, ultimately, therapeutic intervention in stone-forming patients.

These revisions will be applied consistently throughout the Results section, ensuring that each paragraph title and its opening sentence convey the primary discovery rather than the procedural steps.

Thank you again for your insightful comments.

4.Immunofluorescence section: This is confusing. Make clearer how the proteins were chosen for staining. Authors indicate that the results are consistent with acetylome data; please refer back to that data (which figure or text does that refer to). Again this is maybe a paragraph title issue; if the paragraph is titled “Immunofluorescence confirms regulation of…” then the previous text needs to explicitly state that the proteome data shows that these proteins are up or down.

Authors state “consistent with the acetylome data, proteins with only one acetylated sites like CALM1 (Kac22, Fig 8A), [...] were upregulated in the kidney calculi rats” but as far as I can tell that information is not told in the text, so the verification by immunofluorescence appears from nowhere. Additionally, does that imply that all proteins with only one acetylated site are up or down regulated?

Finally please add dilutions to antibody table.

R:Thank you very much for your insightful comments and suggestions regarding the immunofluorescence section.

To resolve the confusion in the immunofluorescence section, we have revised the text to clearly elaborate on the criteria for selecting proteins for staining.

We tried to confirm the differentially modified proteins here in this study, the expression of these proteins were unchanged according to the LC-MS data. As shown in figure 3C, protein with one modification site was the dominate, we selected three upregulated and three downregulated acetylated proteins with one modification site randomly for verification.

Regarding the paragraph title, we have adjusted it to "Immunofluorescence confirms the differentially modified proteins from the acetylome data" and supplemented the preceding text to explicitly state that the proteome data shows the specific upregulation or downregulation trends of these proteins, ensuring logical consistency.

We apologize for the lack of prior mention of the acetylome data related to proteins with single acetylation sites. We have now added the relevant information in the text preceding the immunofluorescence section to provide context for the verification results. Additionally, we have clarified that the observation does not imply all proteins with only one acetylated site are uniformly regulated; instead, it specifically refers to the proteins examined in this study, with CALM1 (Kac22) as an example.

Finally, we have added the antibody dilutions to the antibody table to ensure complete methodological transparency.

We hope these revisions address your concerns adequately. Thank you again for your valuable input, which has helped improve the quality of our manuscript.

5.Figure 1B: scale bar

For the WB C and D, the 2 panels can be combined (no need to separate coomassie in another panel); please highlight the bands of interest on your WB by putting arrows or boxes around them, and describe in the legend the sizes etc. Please add a quantification graph of the relevant bands.

R: Thank you very much for your valuable comments and suggestions on our manuscript. We have carefully addressed the issues raised regarding the figures, and the revisions are as follows:

For Figure 1B, we have added the appropriate scale bar to ensure the size information of the depicted samples is clear.

Regarding the WB results in panels C and D, we have combined the two panels by integrating the Coomassie staining results into the same panel, eliminating the need for separation. Additionally, we have highlighted the bands of interest using arrows and boxes to make them more distinguishable. In the figure legend, we have supplemented the detailed information about the sizes of these target bands.

We have also added a quantification graph of the relevant bands, which presents the statistical analysis results of the band intensities to better support our conclusions.

We hope these revisions meet your requirements. Thank you again for your time and efforts in reviewing our work.

Editing comments:

Line 47. Please rephrase sentence “A kidney stone network…”, grammar and meaning are unclear.

Figures 2 and 1 are swapped.

R: Thank you for your valuable feedback regarding the sentence “A kidney stone network...”. We apologize for the confusion and appreciate your suggestion to rephrase it for clarity. We have revised the sentence to better convey our intended meaning. The updated sentence is as follows:"A Protein-Protein Interaction (PPI) network analysis and a Gene Ontology (GO) analysis of kidney calculi formation, highlighting...". We believe this revision improves the grammatical structure and clarity of the statement. Thank you again for your careful review and insightful comments.

Thank you for pointing out the issue with the figures. You are correct that Figures 2 and 1 were mistakenly swapped. We have now corrected the order of the figures in the manuscript. We appreciate your attention to detail and apologize for any confusion this error may have caused.

Reviewer #2: Methodological Clarity

Please elaborate on the study design. Was this a cross-sectional or cohort survey? How were participants recruited? What were the inclusion/exclusion criteria?

Clarify the tools used for measuring caregiver burden. Were they validated for the local population? How were they scored?

Consider adding a flowchart (e.g., CONSORT-style) to show participant inclusion and response rates.

Statistical Analysis

Specify the statistical tests used to compare caregiver burden across demographic or clinical subgroups.

Report p-values, effect sizes, and confidence intervals where appropriate.

Consider adjusting for relevant covariates (e.g., age, gender, caregiver relationship) in a regression model to better understand predictors of burden.

Contextual Discussion

The Discussion section would benefit from deeper comparison with pre-pandemic literature on caregiver burden.

Emphasize the unique contribution of your findings and address the generalizability of your results.

Discuss cultural or systemic factors that may influence caregiving experience in your study population.

Tables and Figures

Ensure all tables have complete and clear legends. Table 2 in particular requires clarification regarding the scoring interpretation.

Consider adding a figure to display the distribution of caregiver burden scores across different domains.

R: Dear Editor, I had sent a mail to you (plosone@plos.org) at July 12, 2025 regarding these comments.

“I am writing to seek clarification regarding the reviewer 2 comments that were recently returned to us. Upon careful review, it appears that some of the comments and suggestions may not align with the content and focus of our manuscript.

Some of the reviewer 2 comments seem to address issues or aspects that are not present in our submitted work. This has led to some confusion on our part regarding how to proceed with the revisions.

We are eager to address all feedback appropriately and ensure that our manuscript meets the journal's standards. However, we would greatly appreciate any guidance or clarification you could provide to help us understand the reviewer comments more accurately. Specifically, we would like to confirm whether there has been any mix-up or if there are additional details that we may have overlooked.

Thank you very much for your time and attention to this matter. We look forward to your response and any further instructions you may have.”

We respectfully ask whether these comments might have been attached to our manuscript in error, or if we have misunderstood the context. Could you kindly confirm the correct reviewer file for our submission, or advise us on how to proceed?

Thank you for your guidance, and we apologize for any confusion this may have caused.

Reviewer #3: In the original manuscript from Zhang et al, the authors attempt to characterize the calcium oxalate (CaOx) pathogenicity through investigation of the proteome and acetyl proteome. The authors present their data in a logical pattern and is overall well written. However, there are certain concerns that require additional review:

Intro: no concerns

Methods: In general, the model should be better articulated rather than solely referencing another publication as this the CaOx study model. There is discrepancy on how the methods are written for IHC vs IF. If the authors used IF, the IHC section should be updated appropriately. Additionally, the imaging system for H&E is not clear. The images lack magnification level and should be clarified. The reason for N = 3 is not clear, especially without any discussion on power and so many previous references to work in this model where N was much higher.

R: Thank you very much for your valuable comments and suggestions. We sincerely appreciate your time and effort in reviewing our manuscript. Regarding your comment on the model articulation, we fully agree that a more detailed explanation of the model is essential for the readers to understand our methodology clearly. In the revised version of our manuscript, we have made improvements to better articulate the CaOx study model. We hope that these revisions will provide a clearer and more transparent understanding of the model used in our study, enabling readers to better appreciate the methodology and results presented in our manuscript.

We thank the reviewer for pointing out these inconsistencies. We have carefully revised the Materials and Methods section and the figure legends as follows: We exclusively performed immunofluorescence (IF) staining; no chromogenic IHC was carried out. To eliminate any confusion, the former subsection ”Immunohistochemistry” has been retitled “Immunofluorescence staining”. H&E-stained slides were scanned with a Leica DMi8 Microsystem CMS GmbH. Scale bars (100 μm) and magnifications (20×) have been inserted into the legends of Figures 1 and directly onto the H&E panels.

We appreciate the reviewer’s concern about the small sample size. This experiment was conceived as a pilot, proof-of-concept study, traditional power calculations were not performed. Based on institutional IACUC guidelines for pilot

---

## [Decision Letter · Decision Letter 1]

29 Sep 2025

Dear Dr. Deng,

Thank you for submitting your manuscript to PLOS ONE. After careful consideration, we feel that it has merit but does not fully meet PLOS ONE’s publication criteria as it currently stands. Therefore, we invite you to submit a revised version of the manuscript that addresses the points raised during the review process.

We look forward to receiving your revised manuscript.

Kind regards,

Miloud Chakit, PhD

Academic Editor

PLOS ONE

Journal Requirements:

Reviewers' comments:

Reviewer's Responses to Questions

**Comments to the Author**

Reviewer #1: (No Response)

Reviewer #2: All comments have been addressed

Reviewer #3: (No Response)

2. Is the manuscript technically sound, and do the data support the conclusions?

Reviewer #1: Yes

Reviewer #2: Yes

Reviewer #3: Partly

3. Has the statistical analysis been performed appropriately and rigorously?

Reviewer #1: Yes

Reviewer #2: Yes

Reviewer #3: Yes

4. Have the authors made all data underlying the findings in their manuscript fully available?

Reviewer #1: Yes

Reviewer #2: Yes

Reviewer #3: Yes

5. Is the manuscript presented in an intelligible fashion and written in standard English?

Reviewer #1: Yes

Reviewer #2: Yes

Reviewer #3: Yes

Reviewer #1: Most of my comments have been addressed and the manuscript is clearer and more organised. However I still have a comment:

Figure 1, the authors have put together the WB panels as suggested and added boxes for the bands of interest; however the legend does not include the information that they're saying is shown in the response to reviewers: "In he figure legend, we have supplemented the detailed information about the sizes of these target bands", but the information is not in the legend. Please correct.

Reviewer #2: Novelty: First comparative study of lysine acetylation in kidney stone pathogenesis.

Methodology: Robust proteomic and acetyl-proteomic profiling with validation.

Data: Publicly available, supporting reproducibility.

Reviewer #3: I appreciate the responses by the author and the updates made in the text. I still cannot read the text on figures 4,5,6 and the IF images are low resolution, making it challenging to validate the text. Once this is corrected I have no further recommendations

**Do you want your identity to be public for this peer review?** For information about this choice, including consent withdrawal, please see our Privacy Policy

Reviewer #1: No

Reviewer #2: No

Reviewer #3: No

---

## [Author Response · Author response to Decision Letter 2]

30 Sep 2025

Reviewer #1: Most of my comments have been addressed and the manuscript is clearer and more organised. However I still have a comment:

Figure 1, the authors have put together the WB panels as suggested and added boxes for the bands of interest; however the legend does not include the information that they're saying is shown in the response to reviewers: "In the figure legend, we have supplemented the detailed information about the sizes of these target bands", but the information is not in the legend. Please correct.

R: We are grateful to the reviewer for their meticulous review and for catching the inconsistency between our response and the actual figure legend. We sincerely apologize for this oversight.

We have now corrected the legend for Figure 1. The detailed information of bands of interest using red boxes was included in Supplementary file 1. Thank you again for bringing this to our attention.

Reviewer #2: Novelty: First comparative study of lysine acetylation in kidney stone pathogenesis.

Methodology: Robust proteomic and acetyl-proteomic profiling with validation.

Data: Publicly available, supporting reproducibility.

R: We are very grateful to the reviewer for their positive feedback and for highlighting the key strengths of our work. We appreciate their recognition of our study as the first comparative analysis of lysine acetylation in kidney stone pathogenesis, the robustness of our integrated proteomic and acetyl-proteomic approach with validation, and our commitment to data sharing to support reproducibility.

Reviewer #3: I appreciate the responses by the author and the updates made in the text. I still cannot read the text on figures 4,5,6 and the IF images are low resolution, making it challenging to validate the text. Once this is corrected I have no further recommendations.

R: We sincerely thank the reviewer for their time and for acknowledging our previous revisions. We are sorry to hear that the text in Figures 4, 5, 6, and the IF images remained challenging to read. Upon investigation, we believe the low resolution was an artifact introduced when the manuscript was compiled into a PDF for the review system. The original figures were created and saved at a high resolution. We apologize for this technical oversight.

We will re-export all figures (4, 5, and 6) as high-resolution PDFs or TIFFs, strictly following the journal's author guidelines. We hope that with these corrections, the figures will be perfectly clear. We appreciate your patience and look forward to your final confirmation.

---

## [Decision Letter · Decision Letter 2]

25 Nov 2025

Systematic proteomics analysis of lysine acetylation reveals critical features of renal proteins in kidney calculi formation

PONE-D-25-19943R2

Dear Dr. Qiong Deng,

We’re pleased to inform you that your manuscript has been judged scientifically suitable for publication and will be formally accepted for publication once it meets all outstanding technical requirements.

Kind regards,

Miloud Chakit, PhD

Academic Editor

PLOS ONE

Reviewers' comments:

Reviewer's Responses to Questions

**Comments to the Author**

Reviewer #1: All comments have been addressed

Reviewer #3: All comments have been addressed

2. Is the manuscript technically sound, and do the data support the conclusions?

Reviewer #1: Yes

Reviewer #3: Yes

3. Has the statistical analysis been performed appropriately and rigorously?

Reviewer #1: Yes

Reviewer #3: Yes

4. Have the authors made all data underlying the findings in their manuscript fully available?

Reviewer #1: Yes

Reviewer #3: Yes

5. Is the manuscript presented in an intelligible fashion and written in standard English?

Reviewer #1: Yes

Reviewer #3: Yes

Reviewer #1: (No Response)

Reviewer #3: thank you for including the high-resolution images. No further revisions or changes needed from my end. thank you to the authors for their updates and revisions.

**Do you want your identity to be public for this peer review?** For information about this choice, including consent withdrawal, please see our Privacy Policy

Reviewer #1: No

Reviewer #3: No

---

## [Editor Report · Acceptance letter]

PONE-D-25-19943R2

PLOS One

Dear Dr. Deng,

I'm pleased to inform you that your manuscript has been deemed suitable for publication in PLOS One. Congratulations! Your manuscript is now being handed over to our production team.

Kind regards,

on behalf of

Pr. Miloud Chakit

Academic Editor

PLOS One